# Development of a Large Flood Regionalisation Model Considering Spatial Dependence—Application to Ungauged Catchments in Australia

**Khaled Haddad [1],\* and Ataur Rahman [2]**

[1]  Environment and Infrastructure Division, Cumberland Council, Merrylands, NSW 2170, Australia
[2]  School of Computing, Engineering and Mathematics, Western Sydney University; Locked Bag 1797, Penrith NSW 2750, Australia; a.rahman@westernsydney.edu.au
\*  Correspondence: khaled.haddad@cumberland.nsw.gov.au; Tel.: +61-470-366-482

**Abstract:** Estimation of large floods is imperative in planning and designing large hydraulic structures. Due to the limited availability of observed flood data, estimating the frequencies of large floods requires significant extrapolation beyond the available data. This paper presents the development of a large flood regionalisation model (LFRM) based on observed flood data. The LFRM assumes that the maximum observed flood data over a large number of sites in a region can be pooled together by accounting for the at-site variations in the mean and coefficient of variation. The LFRM is enhanced by adding a spatial dependence model, which accounts for the net information available for regional analysis. It was found that the LFRM, which accounts for spatial dependence and that pools 1 or 3 maxima from a site, was able to estimate the 1 in 1000 annual exceedance probability flood quantile with consistency, showing a positive bias on average (5–7%) and modest median relative errors (30–33%).

**Keywords:** large floods; spatial dependence; Generalized Extreme Value; regional flood frequency analysis; ungauged catchments

---

## 1. Introduction

The estimation of rare to very rare floods is needed for many engineering applications, such as planning and designing large hydraulic structures, dam spillways, and flood control levees. Due to the limited availability of observed flood and rainfall data, the estimation of rare to very rare flood frequencies remains a challenging task. All methods used to estimate rare to very rare floods involve significant extrapolation beyond recorded flood and rainfall data. The term 'rare' flood(s) refers to floods with annual exceedance probabilities (AEPs) of 1 in 50 to 1 in 100 [1]. Floods in the AEP range from 1 in 100 to the 'credible limit of extrapolation' (AEP in the order of 1 in 2000) are referred to as 'very rare' floods, while floods from the credible limit of extrapolation to the probable maximum flood (PMF) are termed 'extreme' floods. Due to knowledge and data limitations and the uncertainty involved in extrapolating beyond available data, the errors in final estimates can be quite high. The average size of recorded flood data series for Australian small to medium sized catchments is about 33 years [2].

To make better use of the available flood data and to be able to transfer this information to ungauged catchments, regional estimation methods are used, such as the index flood method [3,4]. The basic idea is that if a region is relatively homogenous, then the estimation of large to rare flood quantiles at a given site may be improved by using the larger observations at other sites as well (i.e., a trade-off between space and time). Some studies both in the past and present, and on an international scale, have looked at the advantages and disadvantages of different regional models for rare, very rare, and extreme floods (e.g., [5–11]).

A new probabilistic model (PM) was introduced by [11] specifically for this sort of analysis. Majone and Tomirotti [11] originally calibrated the PM for Italian rivers, and extended the method using 7300 historical series of annual maximum flows observed at gauging stations belonging to different geographical areas around the world. This model is based on the assumption that the standardised maximum values ($Q_{max}$) of the annual maximum flood series (AMFS) from a large number of individual sites in a region are independent and can be pooled. The PM concept is identical to the basic concept of station-year methods: observed data from an assumed homogenous region are pooled and a non-parametric flood frequency curve is fitted on a probability plot. The traditional approach in the station-year methods is to achieve an acceptable degree of homogeneity within the region by standardising by the at-site mean or median values. The novel aspect of the PM's standardisation is to take into account not only the at-site mean, but also the at-site coefficient of variation (CV) values of the time series data. This unique form of standardisation allows the pooling of more data from many stations compared to the standard index methods. It is known that the station-year method suffers from problems associated with inter-site dependence. In the PM technique presented by Majone and Tomirotti [11], it was assumed that the individual values in the standardised and pooled data series are independent. This assumption may be valid if the data being pooled comes from stations that are spread over a very large region, as with Majone and Tomirotti [11].

The 'large flood regionalisation model' (LFRM) described by [12] is a modified version of the PM technique that was applied to a large set of catchments in Australia. Detailed examination showed that the values in the pooled LFRM data series used in this study tended to cluster in some years, with very few events in other years. This appears to violate the assumption of independent distribution of the events in time and indicates that some of the events occurring in the same year might have resulted from the same hydro-meteorological events. The testing of the LFRM by [12] has demonstrated that if the Australian LFRM data series is assumed to be independent, the LFRM tends to underestimate the at-site flood frequency estimates.

On the basis of these findings from the initial application, the LFRM was further developed by (i) coupling it with a spatial dependence model ([13,14]) that reflects the reduction in the net information due to spatial dependence (e.g., [15–20]); (ii) pooling more data by taking the top 3 maximum values in a region; and (iii) combining it with Bayesian generalised least squares (BGLSR) [21] and the region of influence (ROI) approach to develop regional prediction equations, so that the LFRM can be applied to ungauged catchments. In the ROI approach, a separate region can be formed for each of the gauged catchments by drawing an appropriate number of nearby stations. The advantages of the ROI approach are discussed in more detail in [21] and [13,22,23].

Points (i), (ii), and (iii) above are, in essence, the main innovations of the LFRM model being presented in this paper. An advantage of the LFRM proposed in this paper is that it offers an alternative to traditional approaches of rare flood estimation methods based on rainfall runoff models, where time and resource constraints may not permit the development of detailed rainfall-based methods. Moreover, there is no guarantee that rainfall-based methods provide the best possible estimates.

The remainder of the paper is organised as follows: Section 2 presents the data used in the development of the LFRM and the spatial dependence model. The methodology, concept, and results of the spatial dependence models are given in Section 3. In Section 4, the results of the LFRM and the results of the enhanced LFRM in the light of spatial dependence are also provided. Section 4 also details the results associated with the use of the enhanced LFRM for ungauged catchment estimation. Conclusions are drawn in Section 5.

## 2. Data Description

A total of 654 gauging stations in Australia with reasonable record lengths (19 to 96 years with a mean of 34 years) suitable for regional flood frequency analysis (RFFA) were assembled as a part of Australian Rainfall and Runoff (ARR) revision 'Project 5 Regional Flood Methods' [2]. The selected 654 sites are shown in Figure 1. The streamflow data of these sites were prepared following stringent

procedures, as described in [24]. Streamflows at these sites are essentially unregulated and have not been affected by major land use changes. The catchment areas range between 0.1 and 7406 km$^2$ (mean: 350 km$^2$). One does expect that the useful information for RFFA increases with the increasing number of stations in the region; however, the net information does not increase proportionally with the increasing number of stations within a given region, due to spatial dependence between data at gauging stations. While the shorter record lengths in this study (<25 years) would introduce notable uncertainty in parameter estimation for at-site frequency analysis, they were included, as they still contain useful additional information for the pooled data set of large flood events. From the 654 stations, two datasets for the LFRM were established: (a) 626 stations with reasonable concurrent record lengths were selected for use in the development of the LFRM and constant spatial dependence model. (b) The remaining 28 stations (6 from each of the states of New South Wales (NSW), Victoria (VIC), Queensland (QLD), Western Australia (WA), and 4 from Tasmania (TAS)) located around Australia were put aside for testing and validation with the LFRM. All 626 catchments were used to develop the prediction equations for the model parameters (i.e., mean and CV) of the LFRM (for ungauged catchments) using the BGLSR and the ROI approach.

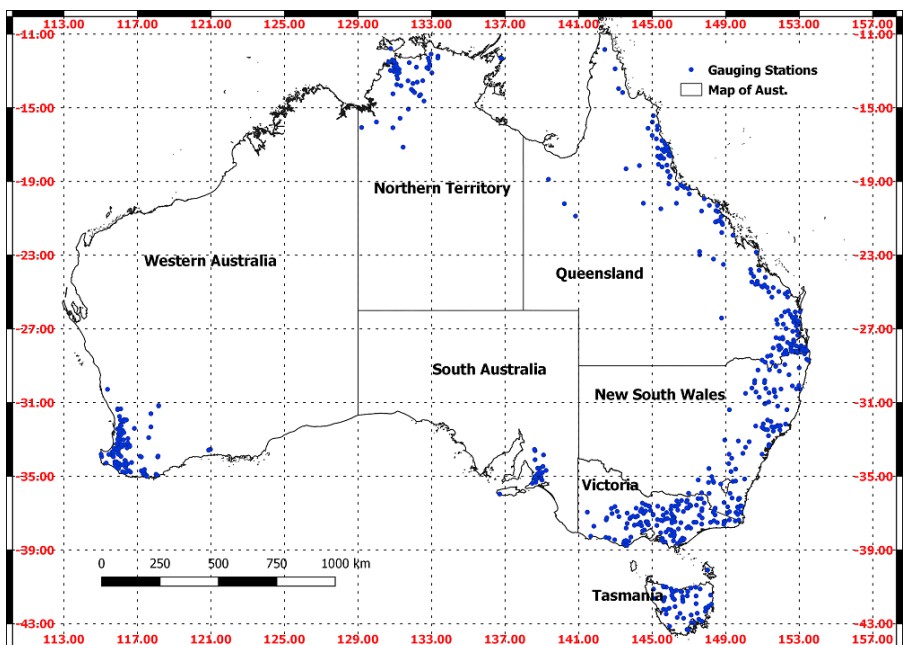

**Figure 1.** Geographical distribution of the selected 654 stations from all over Australia.

With AMFS data, large errors are often associated with the highest flows in the data set because of the nature of the rating curve extrapolation errors. As will be discussed in the methodology section, the LFRM uses the largest 1 to 3 observed maxima values from each station in the region. Therefore, any errors in these observations can introduce significant error into the LFRM final quantile estimates. As can be read in [24], a rating ratio check was introduced and used to cull stations with significant rating curve error.

## 3. Methods

### 3.1. Identification of A Suitable Parent Distribution

The LFRM concept is primarily non-parametric, and therefore an assumption regarding a particular distribution is not required. However, a probability distribution is fitted to the AMFS in order to derive a generic relationship for the effective number of stations $N_e$, which is used to adjust the plotting position of the LFRM points. By comparing a range of methods and distributions, it was found

that the generalised extreme value (GEV) distribution is quite appropriate to approximate the AMFS in Australia based on (i) L-moments ratio diagram, the L-moment goodness-of-fit test, and (ii) the Anderson Darling goodness-of-fit Monte Carlo test. Based on visual inspections of the standardised (as per index flood) flood frequency curves developed, the GEV appeared to be a good candidate to describe the AMFS data for different Australian states. The GEV distribution seemed to capture the higher flows much better than the other competing three-parameter distributions. Further information and discussion can be found in [25,26].

### 3.2. Estimating Inter-Site Dependence

Spatial dependence can be accounted for through the use of a spatial dependence model, which defines the effective number of independent stations in a region ($N_e$) [13,18,20]. The value of $N_e$ can be calculated from the relative position of two frequency curves, the regional maximum curve (this is formed by pooling the highest values of the standardised maxima from the stations in the streamflow gauging network considered each year), and the regional average curve (which is formed by the average curve for the streamflow network considered, i.e., the average of the standardised at-site curves) (or typical curve) [17]. This can be measured on a Gumbel plot ($Y_T$) by $\ln(N_e)$, the horizontal separation between the two frequency curves, as seen Figure 2. As shown by Dales and Reed [17] that the value of $N_e$ is constant irrespective of quantile or average recurrence interval (ARI) if the annual maxima are GEV distributed and the shape parameters of the regional maximum and regional average distributions are equal. In this study, a relatively simple model of spatial dependence was obtained by ignoring the possible variation of $N_e$ with ARI/annual exceedence probability (AEP).

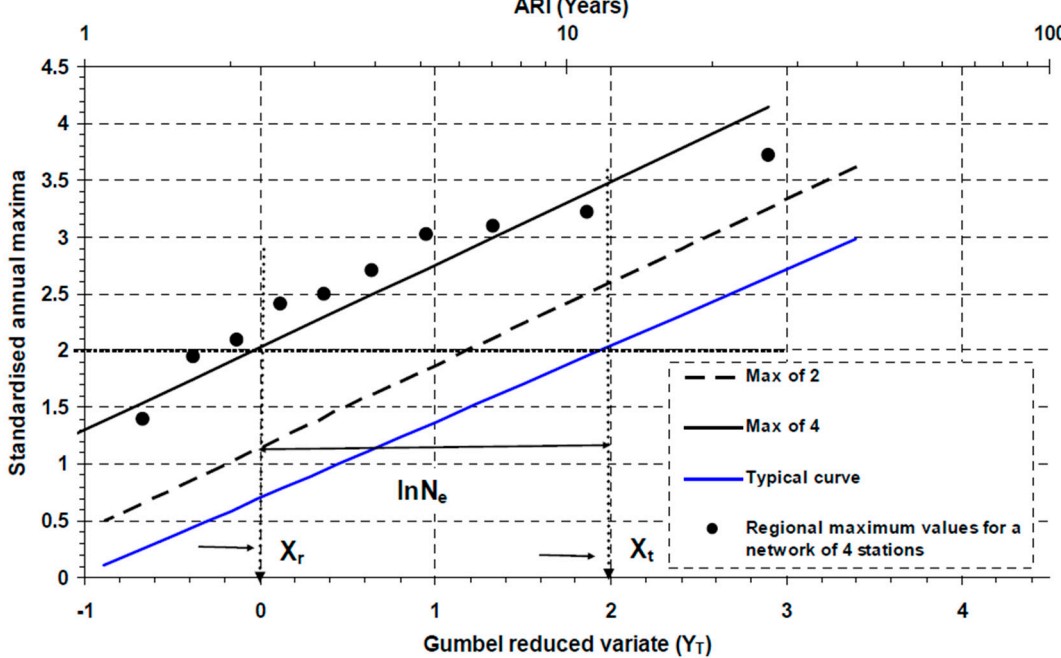

**Figure 2.** Example plot of regional maximum and typical growth curves and the effective number of independent stations on a Gumbel plot for a random network of 2 and 4 gauging sites in Tasmania.

For each regional network of stations, a fixed value of $N_e$ is calculated from the first probability weighted moments ($\beta_o$-mean) $\beta_0^r$ and $\beta_0^t$ of the regional maximum (superscript r) and typical point (superscript t) data:

$$N_e = \left[ \frac{\left( \beta_0^r - x_{bound} \right)}{\left( \beta_0^t - x_{bound} \right)} \right]^{-\frac{1}{\kappa}} \qquad (1)$$

where κ is the shape parameter of the GEV distribution, and $x_{bound}$ is the bound of the typical point GEV distribution [17].

Network sizes of N = 2, 4, and 8 stations were used to determine a relationship between N and $N_e$. Based on concurrent record lengths between sites (average 18 years) a maximum number of 8 stations in a network were adopted for the experiments carried out. To establish an indication of the typical degree of dependence in a network of size N, different forms of constrained and random sampling were adopted, and then $N_e$ was related to the average correlation coefficient (ρ) of concurrent annual maxima at pairs of stations for the particular network size. The different forms of constrained and random sampling were (i) using a ROI approach to pool stations closest to the station of interest using N = 2, 4, and 8, (ii) pooling the closest 20 stations and then randomly sampling N = 2, 4, and 8 sites, and (iii) totally random sampling N = 2, 4, and 8 sites from the region. The different sampling approaches provide different information about ρ. For example, the ROI and random ROI method was useful when investigating small networks which are highly correlated. The total random network samples were over a broader range of correlation values. Further details on this can be seen in [13,14]. This was carried out on the real dataset and on a simulated dataset, with the main purpose of establishing a suitable model to describe the spatial dependence in each region/state and then combining the results to form one relationship to use for all of Australia. For the simulated data, the multi-site maxima were generated according to [20] using a GEV distribution. For the generated dataset, the constant correlation coefficient varied from 0 to 0.5 in steps of 0.1. The overall steps in the analysis carried out in this paper are summarised in a flowchart given in Figure 3.

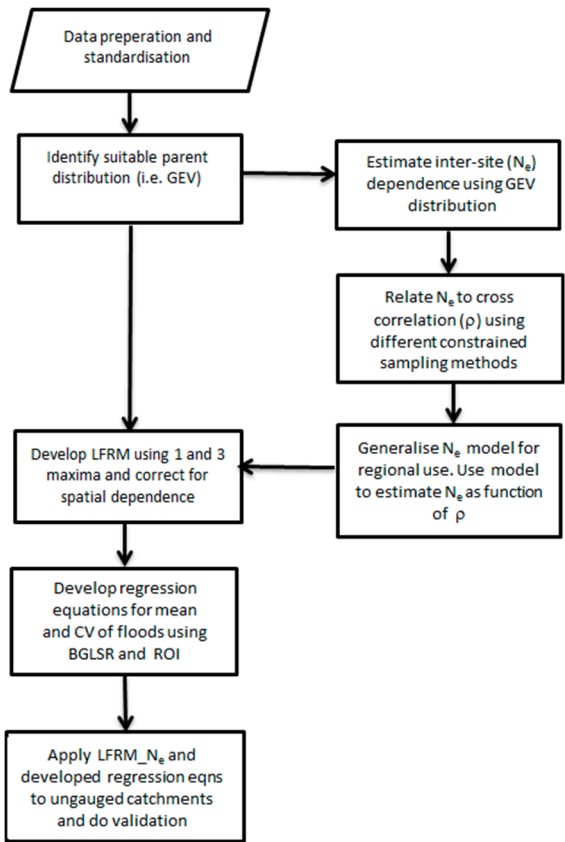

**Figure 3.** Flowchart with the different methods described in the methodology.

### 3.3. Establishing and Generalising $N_e$

The $N_e$ values were obtained for these different network sizes by fitting the mean as described in Section 3.2 for the real and simulated datasets for the different networks and regions. It was found that

the total random network exhibits less spatial dependence than both the ROI and random ROI networks. Indeed, sites that are closer together are more likely to show more spatial dependence. Importantly, the same features as detailed above were seen with the simulated data; however, the simulated data showed less spatial dependence in the 'total random network' as compared to the real dataset.

To avoid regional variations in the derived spatial dependence model, a regional approach was used to combine all the experimental results together for all of Australia. Regression analysis was used to relate $N_e$ to the average correlation coefficient ($\rho$) of concurrent AMFS at pairs of stations for the different networks and regions for each of the adopted experiments (this includes the real data and simulated data). To derive the regression equation, it was determined to be more appropriate to build a general model that relates the ratio $LN(N_e)/LN(N)$ to the average correlation coefficient ($\rho$), similar to that of Dales and Reed [17].

The form of the constant $N_e$ model is given by Equation (2) which was calibrated by combining all the models for each of the Australian states into one generic equation. The final form of Equation (2) was identified by investigating the real and simulated data sets:

$$\frac{LN(N_e)}{LN(N)} = a + b\overline{\rho} \tag{2}$$

Overall, the one variable model (see Equation (2)) provided a relatively good fit for the experimental data. The fitted parameters of the constant $N_e$ model for Australia (overall) are given in Table 1 for the real and simulated datasets. The final parameter values for the general Australian spatial dependence model were found by combining the different network values of the ratio $LN(N_e)/LN(N)$ and developing a regression equation of the form represented by Equation (2), then taking the average of the coefficient values of the developed regression equations.

**Table 1.** Properties of the constant $N_e$ spatial dependence model.

| Region | Real Data | | | Simulated Data | | |
|---|---|---|---|---|---|---|
| | a | b | $R^2$ | a | b | $R^2$ |
| Australia | 1 | −0.66 | 88 | 1 | −0.63 | 99 |

The coefficient of determination ($R^2$) values for the final models (see Table 1) fitted to the real and simulated data sets are quite high, suggesting that the use of the constant $N_e$ model should result in improved $N_e$ estimates compared to the values calculated directly from the AMFS data in each station network.

The comparison of the fitted $N_e$ values for the real data computed using Equation (1) and those by the spatial dependence Equation (2) are shown in Figure 4. The figure below illustrates that the scatter in the spatial dependence model estimates increases with increasing N. The scatter may also be attributed to natural and sampling variability from site to site, given that the concurrent record length for analysis was very modest. Figure 4 and Table 1 show the overall satisfactory performance of Equation (2), as far as practical application is concerned.

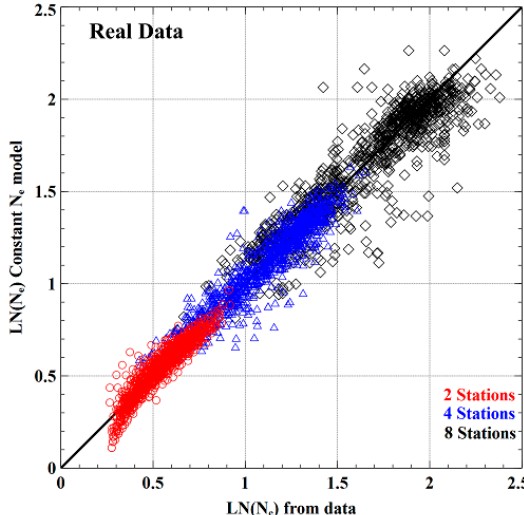

**Figure 4.** Comparison of directly computed $N_e$ from the annual maximum flood series (AMFS) data and $N_e$ by the constant $N_e$ model. The black line represents a 1:1 line.

## 4. Results

### 4.1. Development and Calibration of LFRM for Australian Data

The selected $Q_{max}$ (1 and 3) (i.e., the top 1 and 3 maximum data points from each station's AMFS data, referred to as $Q_{max}$), are first standardised by the at-site average of the AMFS data (mean), and then plotted in the (CV, $Q_{max}$/mean) plane. Figure 5 shows such a plot for (1 and 3 max) the study data set, consisting of 626 data points (1 max), 1878 data points (3 max) from 626 sites, which suggests the following relationship:

$$\frac{Q_{max}}{mean} = c + \alpha CV^{\psi} \tag{3}$$

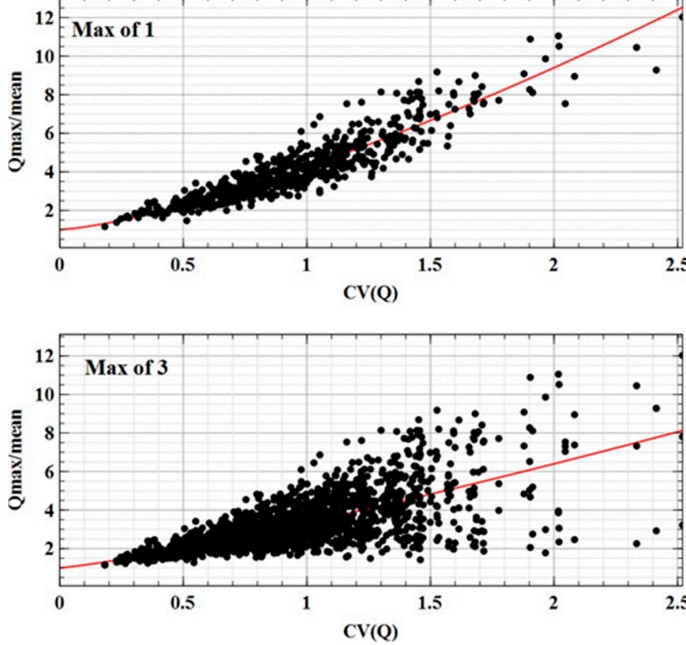

**Figure 5.** Scatter of $Q_{max}$/mean data in the (CV(Q), $Q_{max}$/mean) plane, and non-linear interpolation function, which is represented with a red line. (Top panel) Maxima 1; (bottom panel) Maxima 3.

The coefficients (c, $\alpha$, and $\psi$) of Equation (3) were estimated by the maximum likelihood method for each of the plots in Figure 5. The estimated coefficients, along with their $R^2$ values, are provided in Table 2.

**Table 2.** Coefficients of non-linear interpolation from Figure 5.

| $Q_{max}$-AMFS | c | $\alpha$ | $\psi$ | $R^2$ (%) |
|---|---|---|---|---|
| 1 | 1 | 3.25 | 1.37 | 87 |
| 3 | 1 | 2.35 | 1.20 | 70 |

The $R^2$ values in Table 2 suggest that the estimated coefficients provide a reasonably good fit to the experimental data; this is more evident, however, when pooling the top 1 AMFS. When pooling 3 top maxima, a greater scatter is noticed, as can be seen in Figure 5; this is also supported by the drop in $R^2$ value. An important note is made here on whether the weaker relationship with CV is compensated for later on by having additional data points to define the lower end of the distribution. What can be observed from Table 2 is that the exponent $\psi$ is appreciably greater than unity (as would be the case for a Gumbel distribution for 1 maxima) and decreases slightly with the pooling of more data (i.e., 3 max).

Based on Figure 5, and assuming that a large part of the scatter can be explained by variations in the average recurrence interval (ARI/AEP) of the AMFS data, the best way to model the scatter is to search for a LFRM function in the form of:

$$\frac{Q_{max}}{mean} = c + f(ARI)CV^{\psi} \tag{4}$$

where it is assumed that $f(ARI/AEP)$ is a function of the ARI/AEP only, and can be substituted for the coefficient $\alpha$. From Equation (3), the calibration procedure is based on the introduction of a new standardised variable, which can be defined by:

$$Y_{max} = \frac{\left(\frac{Q_{max}}{mean}\right) - c}{CV^{\psi}} \tag{5}$$

where c and $\psi$ are based on the coefficients according to the number of annual maxima pooled (e.g., 1 or 3 maxima). This form of standardisation (Equation (5)) takes into account not only differences in the mean values, but also of the CV, raised to the power appropriate for a specific regional data set. As expected, as a result of this new standardisation, $Y_{max}$ was practically uncorrelated with the CV, as was confirmed by the very small $R^2$ of 0.0037 referring to the same set of data points for using the top 3 annual maxima. The following plotting position formula (Equations (6)–(8)), proposed by Majone and Tomirotti [11], was applied to estimate the ARI or the empirical non-exceedance frequency (AEP) of each of the $Y_{max}$ values in the pooled data sets (i.e., max of 1 and 3) from the N = 626 sites. In order to define the form of the distribution of the variable $Y_{max}$, the top 1 and 3 annual maxima values of each site's data were used. Here, the major assumption made is that the *i*th value of the $Y_{max}$ series is independent of the other values and that the $Y_{max}$ values belong to the same population. It follows that the plotting position of the $Y_{max}$ can be provided by the following equations (Majone and Tomoirotti [11]):

$$P(Y_{max} \leq y_{max}) = P\left(Y \leq y_{max}\right)^{n_a} \tag{6}$$

where Y is the at-site standardised annual maximum and $n_a$ is the site sample size (taken as the average of the site samples sizes, which is 34 for this study). Now, sorting the pooled sample of standardised maxima consisting of N = 626 (and L = 626 or 1878) in decreasing order and define $y_{max}$ (m) as the mth ranked value in the pooled sample. The ARI of $y_{max}(m)$ (expressed as T years) can be estimated using

$$\frac{m}{N} = 1 - P\left(Y_{max} \leq y_{max}(m)\right) = 1 - P\left(Y \leq y_{max}(m)\right)^{n_a} = 1 - \left(1 - \frac{1}{T}\right)^{n_a} \tag{7}$$

Rearranging leads to:

$$T = \cfrac{1}{\left(1 - \left(1 - \frac{m}{N}\right)^{\frac{1}{na}}\right)} \tag{8}$$

From this definition, the estimated ARI/AEP values would ideally be assumed to be representative of actual return periods. However, this may not be the case for the Australian flood data set, as many of the gauging sites used here are very close together spatially (see Figure 1) and hence there would be significant inter-site dependence. The plot of $Y_{max}$ vs. $Y_T$ (where $Y_T$ is the Gumbel reduced variate and is used as a surrogate for ARI or AEP), where $Y_T = -\ln[-\ln(1-1/T)]$ is shown in Figure 6 for $Y_{max}$ (L = 626 and 1838). The plots for L = 626 and L = 1838 sites in Figure 6 (bottom curves for all two plots) are in line with what would be expected from using the additional data points. Clearly, the impact of using a greater number of maxima, e.g., 3 maxima, seems to provide a very smooth empirical distribution that is fitted closely by the distribution function. The plots also reveal that the experimental data can be approximated by a second-degree polynomial function of $Y_T$ as given by Equation (9), whose model coefficients and $R^2$ values can be seen in Table 3 for the different pooling of the annual maxima (i.e., top 1 and 3 maxima):

$$Y_{max} = C_1(Y_T)^2 + C_2(Y_T) + C_3 \tag{9}$$

which, in terms of $Q_{max}$/mean, takes the following form

$$\frac{Q_{max}}{mean} = c + \left(C_1(Y_T)^2 + C_2(Y_T) + C_3\right)CV^\psi \tag{10}$$

Equations (9) and (10) yield the analytical expression of the LFRM model for the study data, set using the top 1 and 3 annual maxima. The appropriate values of the coefficients in Table 3 are substituted into Equations (9) and (10). However, this formulation does not allow for the effects of the inter-site dependence.

**Table 3.** Coefficients and $R^2$ values of $Y_{max}$ polynomial interpolation from Figure 6 for N and $N_e$ sites.

| $N_e$ sites | $C_1$ | $C_2$ | $C_3$ | $R^2$ |
|---|---|---|---|---|
| 1 | −0.0078 | 0.504 | 2.57 | 0.985 |
| 3 | −0.010 | 0.954 | 0.861 | 0.995 |
| **N sites** | $C_1$ | $C_2$ | $C_3$ | $R^2$ |
| 1 | −0.0263 | 0.787 | 0.52 | 0.997 |
| 3 | −0.053 | 1.13 | −0.603 | 0.999 |

### 4.2. Revision of LFRM for Spatial Dependence

The LFRM for the study data in its current form (see Equations (9) and (10)) does not allow for the effect of inter-site dependence. In this section, spatial dependence is accounted for through the use of the spatial dependence model derived in the previous section (see Equation (2)). For this study, the use and calculation of $N_e$ for application with the LFRM is illustrated. Firstly, the average correlation for each pair of sites was calculated for the region by computing the correlation coefficient from a regional relationship with distance for all of the Australian states. The average correlation coefficient was found to be $\bar{\rho} = 0.26$. Secondly, using Equation (2) along with the coefficients for the Australian spatial dependence model given in Table 1 (using the real and simulated data) and $\bar{\rho} = 0.26$, the $N_e$ was estimated. The calculated $N_e$ value, along with the effective record length, is given in Table 4. From Table 3, it can be seen that the results from the real data match reasonably well with the simulated data.

Using the calculated $N_e$ value of 207 (from the real dataset) in Equations (7) and (8) instead of the total number of stations (N = 626), we can now estimate the new plotting position of the pooled data

points for 1 and 3 maxima. The new interpolated curve for Equations (9) and (10) has new coefficient values. The revised coefficient values of the LFRM have now been corrected for the spatial dependence in the dataset. The appropriate values of the coefficients of Equations (9) and (10) are given in Table 3. Differences are clearly seen in the coefficients of the LFRM when comparing the results of the dataset using N and $N_e$ sites; this is due to the reduction of the total useful information (i.e., the effective number of stations). The new interpolated frequency curves can be seen in Figure 6 (both panels, top curves).

**Table 4.** Total record length (L) and effective record length ($L_e$) for all the Australian datasets.

| Region | N | L | Constant $N_e$ Model—Real Data | | Constant $N_e$ Model—Simulated Data | |
|---|---|---|---|---|---|---|
| | | | $N_e$* | $L_e$ | $N_e$* | $L_e$ |
| Australia | 626 | 21049 | 207 | 6969 | 228 | 7654 |
| | | | (33%) | | (36%) | |

\* $N_e$ values in parentheses are percentages of N.

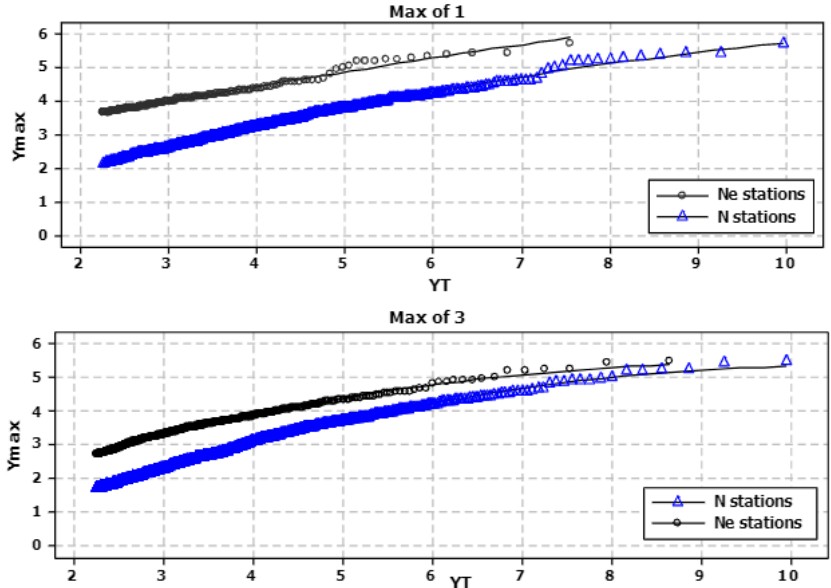

**Figure 6.** Frequency distribution of standardised $Y_{max}$ values using N and $N_e$ stations. (Top panel) Maxima 1; (bottom panel) Maxima 3.

What is striking in Figure 6 is the shift upwards in the frequency curve of the pooled data. Taking the 1 max plot for example, if one compares the $Y_{max}$ value of approximately 4, it can be seen that, if one ignores the spatial dependence, the flood magnitude risk may be notably underestimated (for N sites $Y_{max}$ = 4, AEP = 1 in 87, for $N_e$ sites $Y_{max}$ = 4, AEP = 1 in 8.3). For the pooling of the 1 max and correcting for spatial dependence (see max of 1 plot in Figure 6) it was found that the range of $Y_{max}$ values for which the fitted model (referred to as LFRM_$N_e$ henceforth) might be considered reliable is approximately 2.2 to 7, which corresponds to AEPs of 1 in 10 to approximately 1 in 2000.

Figure 7 shows the behavior of the dimensionless quantiles derived from Equations (9) and (10) for AEPs of 1 in 100, 1 in 500, and 1 in 1000 for all the pooled data, (i.e., 1 and 3 max), and for the estimated quantiles using N and $N_e$. The dimensionless quantiles for the world model (referred to as the PM (world), based on 7300 gauging stations around the world) developed by Majone and Timorotti [11] are also superimposed for comparison. The comparison with the PM (world) curves in Figure 5 indicates that the LFRM_$N_e$ can explain a good amount of the scatter in these plots, as the set of curves (1 in 100 and 1 in 500 AEP curves) for this extended AEP range (including the 1 in 1000-AEP) captures most of the upper part of the points in the pooled data set of the $Q_{max}$/mean values. The flatter slopes in Figure 7 for 3 max (bottom panel), are consistent with what was shown in Figure 5 and seems to reflect

a weaker relationship of $Q_{max}$/mean with CV. Comparison of the curves for max of 1 and 3 for $N_e$ and N seems to indicate that allowance for spatial dependence has a smaller influence on slope. Figure 7 also indicates that the extra data i.e., 3 max provides slightly better definition of the left-hand tail of the distribution (where the top few points in the right-hand tail are mostly common in all 2 data sets (1 and 3 maxima). Further investigation also revealed that the LFRM_$N_e$ can provide reasonably accurate growth curve estimation for CV values in the ranges 0.60–1.60 (approximately 81% (505 out of 626) of the study catchments fall in this range). However, the LFRM_$N_e$ can perform poorly for some catchments, with CV values greater than 1.70.

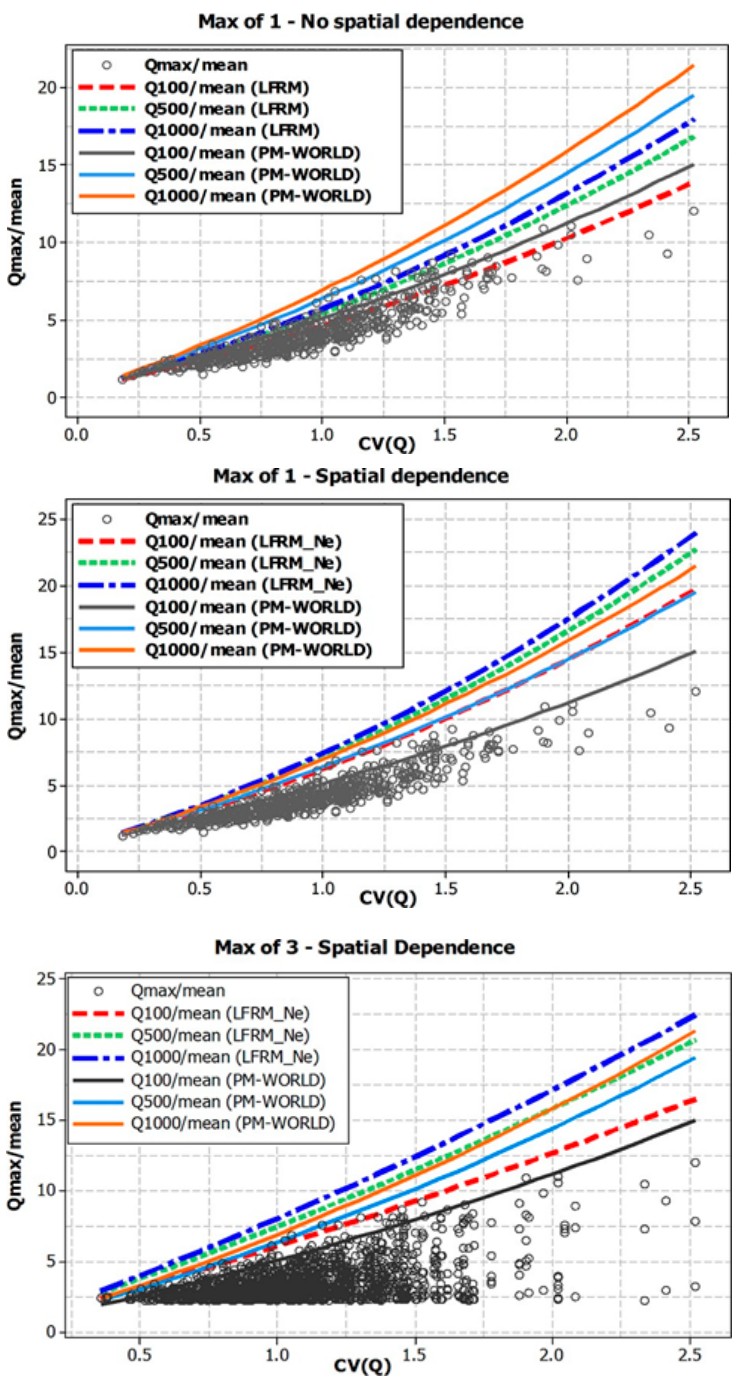

**Figure 7.** Various $Q_{max}$/mean quantiles derived from the LFRM_$N_e$ model and PM (World) model. (Top panel) Maxima 1—no spatial dependence; (middle panel) Maxima 1—spatial dependence accounted for; (bottom panel) Maxima 3—spatial dependence accounted for.

*4.3. Application of the LFRM to Ungauged Catchments*

Our interest is the application of Equations (9) and (10) to ungauged catchments, which requires the estimation of the mean flood and CV for the ungauged catchment in question. The BGLSR and the ROI approach, as discussed in [21], were used to develop the prediction equations for the mean flood and CV of the AMFS data as a function of catchment and climatic characteristics (predictor variables). The prediction equation for the mean flood used a ROI of 30–40 stations, while 65–80 stations were used for the CV, based on the findings from past studies (e.g., [21–23,27]) and which state was being analysed.

The regression equations are presented in general form below:

$$\text{Mean} = \beta_0 + \beta_1(\text{area}) + \beta_2(2I_{12}) \tag{11}$$

$$\text{CV} = \beta_0 \tag{12}$$

The prediction equations developed above using the ROI approach, and Equations (9) and (10) (LFRM_$N_e$ model), were applied to the 28 test catchments, which were not used in developing the prediction equations. To make the comparison more useful and to benchmark the LFRM_$N_e$ model, the developed prediction equations were also used to estimate the mean flood and CV with the PM (world) model developed by Majone and Tomirotti [11]. It must be pointed out however, that the PM (world) model does not contain any of the data used to develop the Australian LFRM. The validation analysis was undertaken for AEPs to 1 in 1000. AEPs in the range of 1 in 50 to 1 in 100 were compared with at-site flood frequency analysis (FFA) (obtained from the fitted log Pearson type 3) distribution using the FLIKE software [28]. Validating beyond the AEP 1 in 100 with at-site FFA estimates was not viewed as reliable, given the very large extrapolation errors involved. Any validation results obtained beyond AEP 1 in 100 would be of little significance for most of the stations.

For the lower AEPs (1 in 500 and 1 in 1000), comparison was made against the results obtained from another regional method where the parameters of the LP3 distribution (i.e., mean, standard deviation, and skew) were regressed against catchment characteristics (known as the PRT—see [21,26] for more details) and flood quantiles were then derived for the 1 in 500 and 1 in 1000 AEPs. The extrapolation of these distributions to the low AEPs also involves a large degree of uncertainty. To assess how well the derived large flood estimates could approximate the observed flood estimates, two numerical measures were applied. Relative bias (BIAS$_r$) was used to assess whether the predicted rare flood quantiles by the LFRM_$N_e$ or PM (world) models systematically under- or overestimated the at-site FFA or the PRT estimates on average, considering all the 28 test catchments.

The relative error values (RE$_r$), with respect to the at-site FFA or the regional parameter regression technique (PRT) estimate, were also obtained. This is by no means the true error of the LFRM_$N_e$ or PM (world) models; the estimated errors represented here by both the BIAS$_r$ and RE$_r$ may be taken as a reasonable indication of consistency of the LFRM_$N_e$ or PM (world) models as compared to FFA and PRT estimates. Here, both the FFA and PRT estimates are associated with a higher degree of uncertainty due to considerable extrapolation involved. It is worth noting here that in calculating the median relative error (RE$_r$), the sign of the relative errors was ignored.

Table 5 summarises the various error statistics with the LFRM_N (i.e., no spatial dependence) and LFRM_$N_e$ models (considering the pooling of 1 and 3 maxima) and the PM (world) model based on the 28 test catchments. If spatial dependence is ignored in the Australian dataset, it is observed that the estimation for the AEP of 1 in 1000 using the LFRM_N model suffers from major underestimation on average (e.g., BIAS$_r$ of −27%) for the ungauged catchment case. Moreover, from Table 5, it can be seen for 1 max and when the pooling of more data is undertaken (i.e., 3 maxima), and spatial dependence (LFRM_$N_e$) is compensated for, the BIAS$_r$ is well corrected. For example, from Table 5, for the 1 in 1000 AEP, the BIAS$_r$ for 1 and 3 max and LFRM_$N_e$ are a 5 and 7% overestimation on average, respectively.

**Table 5.** Summary of error statistics obtained from independent testing associated with the large flood regionalisation model (LFRM) model.

| 1 Max LFRM_N | | | | |
|---|---|---|---|---|
| AEP (1 in Y) | $BIAS_r$ (%) | | $RE_r$ (%) | |
| Model | LFRM_N | World Model | LFRM_N | World Model |
| 1 in 50 | −2 | 12 | 61 | 56 |
| 1 in 100 | −16 | −2 | 66 | 55 |
| 1 in 200 | −18 | 6 | 46 | 33 |
| 1 in 500 | −20 | 5 | 47 | 33 |
| 1 in 1000 | −27 | −1 | 49 | 34 |
| **1 Max LFRM_N$_e$** | | | | |
| AEP (1 in Y) | $BIAS_r$ (%) | | $RE_r$ (%) | |
| Model | LFRM_N | World Model | LFRM_N | World Model |
| 1 in 50 | 40 | 12 | 66 | 56 |
| 1 in 100 | 18 | −2 | 66 | 55 |
| 1 in 200 | 22 | 6 | 28 | 33 |
| 1 in 500 | 15 | 5 | 29 | 33 |
| 1 in 1000 | 5 | −1 | 33 | 34 |
| **3 Max LFRM_N$_e$** | | | | |
| AEP (1 in Y) | $BIAS_r$ (%) | | $RE_r$ (%) | |
| Model | LFRM_N | World Model | LFRM_N | World Model |
| 1 in 50 | 31 | 12 | 58 | 56 |
| 1 in 100 | 14 | −2 | 60 | 55 |
| 1 in 200 | 15 | 6 | 30 | 33 |
| 1 in 500 | 15 | 5 | 31 | 33 |
| 1 in 1000 | 7 | −1 | 30 | 34 |

Focusing on the 3 max results, for the AEPs of 1 in 50 to 1 in 1000-, the $BIAS_r$ values are positive on average for the LFRM_N$_e$, while for the PM (world) models, there are a couple cases of underestimation on average. When compared to the results of preliminary LFRM models (i.e., [12]), the results obtained here present a significant improvement. As found in Haddad et al. [12], the underestimation on average was up to 40%. By pooling more data and also accounting for the inter-site dependence in the LFRM model, the underestimation problem, to a large extent, has been rectified. The results as benchmarked against the PM (world) model are reassuring; this places a higher degree of confidence in the estimates given by the LFRM_N$_e$ model developed here.

The $RE_r$ values in Table 5 show acceptable results, which are comparable to similar regional models for the larger AEP ranges (Rahman et al. [27]). Focusing on the 3 max results, the $RE_r$ values range from 30% to 60% (which are also very comparable to the PM (world) model). It should be noted that in the PM (world) data set most of the stations were so well separated that they were mostly independent of each other, and this was the reason why Majone and Timorotti [11] did not need to work out an effective number of sites. The LFRM_N$_e$ model in this study has refined the approach of the PM (world) model, as significant inter-site dependence exists between stations in the Australian data set.

An error bar plot of the $BIAS_r$ values is given in Figure 8, which displays the central tendency and variability of the sample $BIAS_r$ values over the 28 independent test catchments. Here, Figure 8 displays the mean value (circle symbol) with a 95% limit bar for flood quantiles AEP 1 in 100 to 1 in 1000. While the mean values appear to be different for the two methods (i.e., LFRM_N$_e$ and PM (world) models), the difference is modest because the error bars overlap, suggesting the LFRM_N$_e$ model to be very comparable and even better than the PM (world) model. Moreover, it proves that consistency is achieved for the 3 maxima pooling LFRM_N$_e$ model as the mean values and the spread of $BIAS_r$ values are very similar to the PM (world) model. What is noteworthy is the difference between LFRM_N$_e$ and LFRM_N. The mean values were found to be statistically different, which suggests that

the LFRM_$N_e$ has corrected the negative bias quite well and justifies the use of the LFRM_$N_e$. It is envisaged that as a part of the future assessment of the LFRM_$N_e$, model comparisons will be made against design flood estimates obtained by alternative methods (e.g., spillway design and dam safety studies based on design rainfall-based approaches).

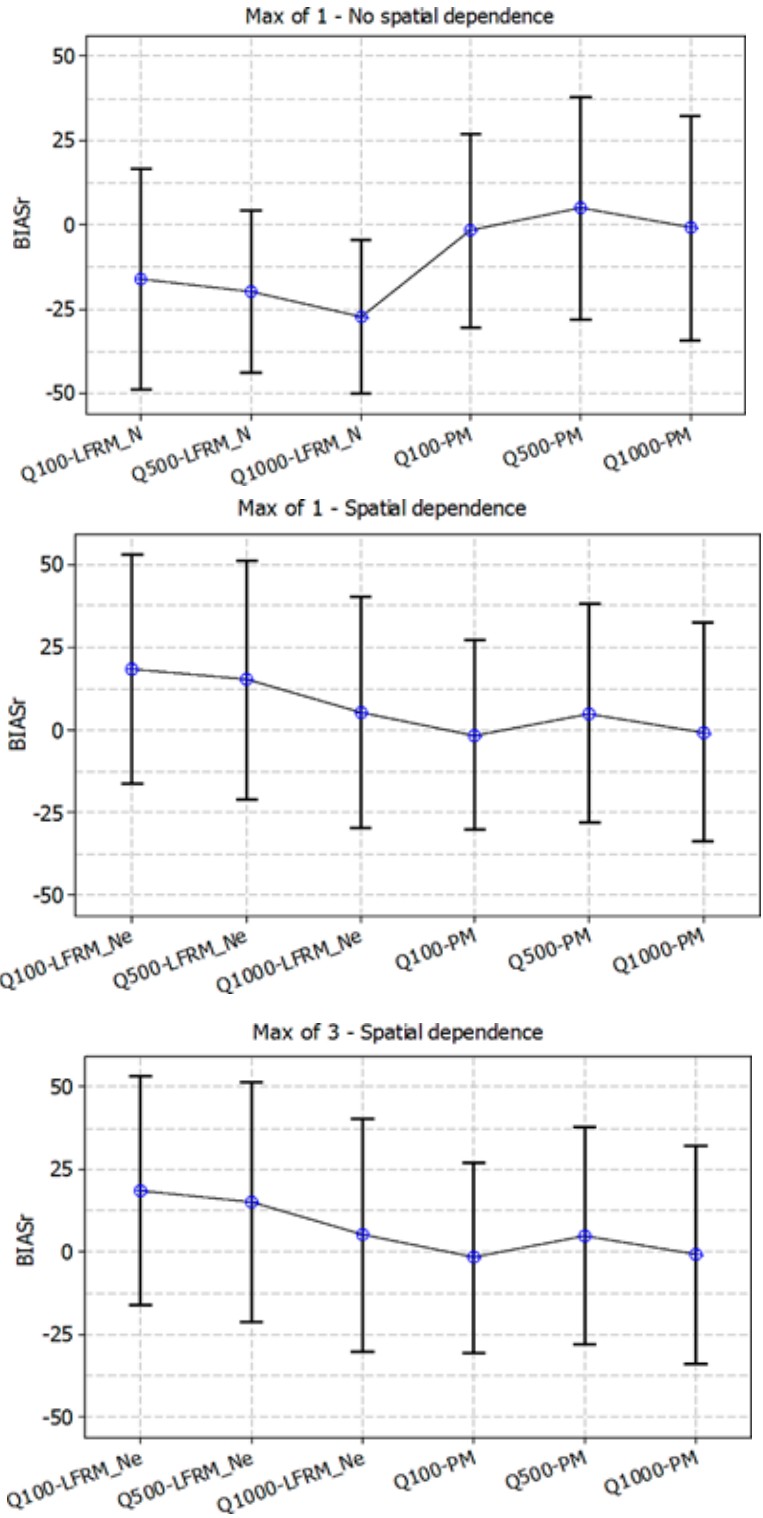

**Figure 8.** Error bar plot of BIAS$_r$ values with the LFRM_$N_e$ and PM (world) models for the 28 test catchments. (Top panel) Maxima 1—no spatial dependence; (middle panel) Maxima 1—spatial dependence accounted; (bottom panel) Maxima 3—spatial dependence accounted for.

## 5. Conclusions

The large flood regionalisation model (LFRM) proposed here can be viewed as an alternative to traditional approaches of large flood estimation, where time and resource constraints may not permit the development of rainfall-based methods. This paper presented the further development and application of a simplified LRFM that pools the top 3 annual maxima flood values from many sites in a region to define the regional curve growth combined with a spatial dependence model for annual maximum flow data. To apply the LFRM to ungauged catchments, Bayesian generalised least squares regression was used to estimate the mean flood and coefficient of variation of annual floods. The extended LFRM coupled with a spatial dependence model offers an alternative method of regional flood estimation for AEPs down to 1 in 1000 years. It has been demonstrated that there is positive bias when estimating the 1 in 1000 AEP flood quantiles. The results obtained in this study represent a step forward for rare to very rare flood estimation for Australian catchments in the absence of detailed flood studies. Further analysis, testing, and future work will also include enhancing the LFRM with the new data being collated for ARR Revision Project 5, comparing results from the LFRM at catchments where detailed flood studies have been undertaken extending to the range of extreme floods (e.g., spillway adequacy assessment), and deriving uncertainty limits to provide a practical and simple tool for rare to very rare flood estimation.

**Author Contributions:** K.H. and A.R. developed the concept. K.H. carried out all the modelling and analysis tasks, while A.R. reviewed the results and also contributed to writing.

**Funding:** This research received no external funding.

**Acknowledgments:** Authors gratefully acknowledge Erwin Weinmann's review of an earlier version of this paper and his helpful comments. The authors would like to acknowledge Australian Rainfall and Runoff (ARR) Project 5 team, in particular, Nanada Nandakumar and George Kuczera for their suggestions on the work presented in this paper. The authors would also like to thank three anonymous reviewers and assistant editor whose comments improved the manuscript notably.

**Conflicts of Interest:** Authors declare no conflict of interest.

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
