# Peer review of "Development of a Large Flood Regionalisation Model Considering Spatial Dependence—Application to Ungauged Catchments in Australia"

_water, doi:10.3390/w11040677_

Round 1

Reviewer 1 Report

The authors present an extension of their previous work on flood frequency analysis methods , to extrapolation of very large floods. The introduction goes straight to the point and cites previous works and limitations of the existing methods. The data and the proposed method are presented clearly. The design of the experiments seems to be correct. The results support the conclusions provided by the authors. The article is suitable to be published in Water.

Author Response

Development of a Large Flood Regionalisation Model Considering Spatial Dependence – Application to Ungauged Catchments in Australia

Submitted to Water – Manuscript ID water-453833

Reviewer

Comments   and suggestions to authors / Reply

Reviewer 1:

The   authors present an extension of their previous work on flood frequency   analysis methods, to extrapolation of very large floods. The introduction   goes straight to the point and cites previous works and limitations of the   existing methods. The data and the proposed method are presented clearly. The   design of the experiments seems to be correct. The results support the   conclusions provided by the authors. The article is suitable to be published   in Water.

Reply:

The authors would like to thank you   Reviewer 1 for his/her constructive comments.

Reviewer 2 Report

The article is more or less unsuccessful attempt to determine the flows of rare floods without proper physical bases. The results are, according to the authors, reasonably good fit or reasonably accurate or quit well. All together, an interesting theoretical approach may be of no practical value. Both graphs on figure 6 are practically identical.

Author Response

Development of a Large Flood Regionalisation Model Considering Spatial Dependence – Application to Ungauged Catchments in Australia

Submitted to Water – Manuscript ID water-453833

Reviewer

Comments   and suggestions to authors / Reply

Reviewer 2:

The   article is more or less unsuccessful attempt to determine the flows of rare   floods without proper physical bases. The results are, according to the   authors, reasonably good fit or reasonably accurate or quit well. Altogether,   an interesting theoretical approach may be of no practical value. Both graphs   on figure 6 are practically identical.

Reply:

Thanks for the comment. The adopted   approach has accounted for the effects of catchment area and rainfall   intensity (physical variables) (see equation 11). The method can be used to   benchmark other available approaches, which is important in hydrologic   modelling applications. Figure 6 is explained in the text as below “The   mean values were found to be statistically different, which suggests that the   LFRM_Ne has corrected the negative bias quite well and justifies   the use of  LFRM_Ne.”

Yes,   the last 2 graphs in Figure 6 should be similar, as they show that pooling 3   maxima LFRM_Ne provides relatively similar results to 1 maxima   LFRM_Ne. This is also noted in the results of the average bias   values of the validation of each method (+5% and +7%, respectively for 1 and   3 max).

Reviewer 3 Report

17-Feb-2019

Comments to Authors:

Manuscript: ‘Development of a Large Flood Regionalisation Model Considering Spatial Dependence – Application to Ungauged Catchments in Australia’

In this work has been formulated a Large Flood Regionalization Model (LFRM) based on observed flood data derived from basins whose areas range between 0.1 and 7,406 km2 in Australia. Unlike other similar works, the LFRM is enhanced by adding a spatial dependence model which accounts for the net information available for regional analysis. The study addresses a quite interesting topic; it offers an alternative method for the estimation of rare to very rare flood frequencies. Furthermore, this is a relevant topic lies within the scope of the MDPI water journal. The article is well organized and neatly written with the appropriate scientific content. Based on the above, I support the publication of this manuscript, but only after a minor revision.

********************************

Title: it fits perfectly the paper content.

Abstract: it is quite adjusted to the paper content, but authors should add some metrics such as the average or median for bias and relative errors (lines 21 and 22).

Introduction: this section provides sufficient background and includes relevant references about some methods used to estimate rare, very rare and extreme floods; particularly at Australia. Objectives clearly stated.

Data description: the description of the data sets is clearly stated. To improve the clarity of Figure 1 authors should use unshaded dots (line 126).

Methods: in my opinion, the research shows a design appropriated and its methods have been adequately described, but for the sake of clarity, I think that this section could be significantly improved if the authors add a flow chart with the different methods described in text, highlighting inputs, applied analysis/procedure, and outputs so that readers could understand this section easier. On the other hand, authors should use a same nomenclature for Ne in lines 133, 147, 154, 159, 168, 171, 191, 193…and others; i.e., subscript ‘e’.    

Line 238: what does ‘black line’ mean? Please, indicate in caption (e.g., the black line represents a 1:1 line).

Results: the results have been presented clearly, but I have some specific comments:

Table 2: authors should add the names of columns 3 and 4 (i.e., ‘alpha’ and ‘psi’).   

Line 271: what does ‘red line’ mean? Please, indicate in caption (e.g., the red line represents the line of best fit).

Line 387: fix this sentence. Figure 5 shows the behavior…

Conclusions: they’re clear and concise and are supported by the results.

Author Response

Development of a Large Flood Regionalisation Model Considering Spatial Dependence – Application to Ungauged Catchments in Australia

Submitted to Water – Manuscript ID water-453833

Reviewer

Comments   and suggestions to authors / Reply

Reviewer 3:

In   this work has been formulated a Large Flood Regionalization Model (LFRM)   based on observed flood data derived from basins whose areas range between   0.1 and 7,406 km2 in Australia. Unlike other similar works, the LFRM is   enhanced by adding a spatial dependence model which accounts for the net   information available for regional analysis. The study addresses a quite   interesting topic; it offers an alternative method for the estimation of rare   to very rare flood frequencies. Furthermore, this is a relevant topic lies   within the scope of the MDPI water journal. The article is well organized and   neatly written with the appropriate scientific content. Based on the above, I   support the publication of this manuscript, but only after a minor revision.

Reply:

The authors would like to thank you   Reviewer 3 for his/her constructive comments.

Reviewer 3:

Title: it fits perfectly the   paper content.

Reply:

Thank you

Reviewer 3:

Abstract:   it is quite adjusted to the paper content, but authors should add some   metrics such as the average or median for bias and relative errors (lines 21   and 22).

Reply:

Metrics have been added in the   abstract in the revised paper

Reviewer 3:

Introduction: this section provides   sufficient background and includes relevant references about some methods   used to estimate rare, very rare and extreme floods; particularly at   Australia. Objectives clearly stated.

Reply:

Thank you

Reviewer 3:

Data   description: the description of the data sets is clearly stated. To   improve the clarity of Figure 1 authors should use unshaded dots (line 126).

Reply:

Updated figure is provided in revised   manuscript.

Reviewer 3:

Methods: in my opinion, the   research shows a design appropriated and its methods have been adequately   described, but for the sake of clarity, I think that this section could be   significantly improved if the authors add a flow chart with the different   methods described in text, highlighting inputs, applied analysis/procedure,   and outputs so that readers could understand this section easier. On the other   hand, authors should use a same nomenclature for Ne in lines 133, 147, 154,   159, 168, 171, 191, 193…and others; i.e., subscript   ‘e’.    

Reply:

Flow chart has been added in revised   manuscript. Same nomenclature for Ne has been   used throughout in the revised manuscript.

Reviewer 3:

Line   238: what does ‘black line’ mean? Please, indicate in caption (e.g., the   black line represents a 1:1 line).

Reply:

Caption has been updated to reflect   black line represents 1:1 line.

Reviewer 3:

Results:   the results have been presented clearly, but I have some specific comments:

Table   2: authors should add the names of columns 3 and 4 (i.e., ‘alpha’ and   ‘psi’).  

Line 271:   what does ‘red line’ mean? Please, indicate in caption (e.g., the red line   represents the line of best fit).

Line   387: fix this sentence. Figure 5 shows the behavior…

Conclusions:   they’re clear and concise and are supported by the results.

Reply:

Names have been added for columns 3   and 4.

Caption has been updated to reflect   red line represents line of best fit.

Sentence has been revised accordingly.

Thank you for the constructive   comments.

Assistant Editor:

*Besides*,   please change the references format (names, year) to [number] and confirm   that all refs should be cited in order.

Reply:

Reference format changed as required.
